# Trends in Hospital Admissions for Mental, Behavioural and Neurodevelopmental Disorders in England and Wales between 1999 and 2019: An Ecological Study

**DOI:** 10.3390/healthcare10112191

**Published:** 2022-10-31

**Authors:** Abdallah Y. Naser, Eman Zmaily Dahmash, Jaber S. Alqahtani, Zahra K. Alsairafi, Fatemah M. Alsaleh, Hassan Alwafi

**Affiliations:** 1Department of Applied Pharmaceutical Sciences and Clinical Pharmacy, Faculty of Pharmacy, Isra University, Amman 11622, Jordan; 2School of Life Science, Pharmacy, and Chemistry, Faculty of Health, Science, Social Care & Education, Kingston University, Surrey KT1 2EE, UK; 3Department of Respiratory Care, Prince Sultan Military College of Health Sciences, Dammam 34313, Saudi Arabia; 4Department of Pharmacy Practice, Faculty of Pharmacy, Kuwait University, Hawalli 13060, Kuwait; 5Faculty of Medicine, Umm Al-Qura University, Mecca 21955, Saudi Arabia

**Keywords:** behavioural, England, hospitalisation, mental, neurodevelopmental, United Kingdom, wales

## Abstract

Objectives: To investigate the trends in hospital admissions for mental, behavioural and neurodevelopmental disorders (MBNDs) in England and Wales. Methods: This is an ecological study using the Hospital Episode Statistics database in England and the Patient Episode Database for Wales. Hospital admission data was collected for the period between April 1999 and March 2019. Results: The most common type of hospital admission was for mental and behavioural disorders due to psychoactive substance use, which accounted for 26.6%. The admission rate among males increased by 8.1% [from 479.59 (95% CI 476.90–482.27) in 1999 to 518.30 (95% CI 515.71–520.90) in 2019 per 1000 persons; *p* < 0.001]. The admission rate among females increased by 0.3% [from 451.45 (95% CI 448.91–453.99) in 1999 to 452.77 (95% CI 450.37–455.17) in 2019 per 1000 persons; *p* = 0.547]. The 15–59 years’ age group accounted for 65.1% of the entire number of such hospital admissions, followed by the 75 years and above age group, with 19.0%. Conclusion: We observed an obvious variation in MBNDs influenced by age and gender. Observational studies are needed to identify other factors associated with increased hospital admission rates related to MBNDs, specifically among the young population (aged 15–59 years) and males.

## 1. Introduction

There is a growing worldwide concern about mental health and well-being. In 2014, a survey regarding adult psychiatric morbidity reported that every week, one in six adults demonstrated symptoms of mental health issues (e.g., anxiety and depression). Furthermore, one in five adults has attempted to take their own life [1]. A report from the Mental Health Foundation in 2016 revealed that almost 50% of adults believed that, in their lifetime, they would experience a mental health problem. Nevertheless, only a third received a diagnosis [1,2]. Worldwide, in 2010, the cost of managing mental health disorders approached $2.5 trillion, with a projected expenditure to exceed $6.1 trillion by 2030 [3]. Indeed, the global disease burden of mental health disorders accounts for 32% of years lived with disability (YLD), which is more than any other condition [4].

In 2012, mental disorders due to substance use accounted for 32.3% of inpatient stays in the USA. Mood disorders were the most common mental disorder, accounting for almost 750,000 stays, while alcohol-related disorders were the most common substance use disorder, accounting for 335,790 stays [5]. Between 2005 and 2014 in the USA, mental health and substance use hospitalisations accounted for 6% of all hospitalisations which showed an increase of 20.1% from 2005 [6].

In 2011, the cost of mental health-related issues to the economy in England has recently been estimated to exceed £100 billion. Treatment costs are projected to double within the next 20 years. Therefore, in response to the burden of mental health, the United Kingdom (UK) government, in 2011, recognised the importance of mental health to the quality of life and produced a strategy to address this issue under the title “No Health Without Mental Health”. The strategy addresses several domains aimed at improving mental health in the UK, which include hospitalisation [7]. However, the need to understand and meet mental health needs is reinforced by growing recognition of the importance of evaluating lifetime trends in hospitalisation due to mental health disorders. In strategy number 3, “More people with mental health problems will have good physical health”, the UK government set key indicators to achieve this objective, including “monitoring the rate of hospital admissions for alcohol-related harm, b) the rate of hospital admissions as a result of self-harm, c) hospital admissions caused by unintentional and deliberate injuries to 5–18-year-olds” [7].

A clinically significant disturbance in a person’s cognition, emotional control, or behavior is indicative of a problem with the psychological, biological, or developmental processes underlying mental and behavioral functioning and characterises mental, behavioral, and neurodevelopmental disorders. Previous studies in the UK have explored the hospitalisation profile for various acute and chronic health conditions [8,9,10,11,12,13,14,15,16,17,18,19]. However, none have explored the hospitalisation pattern for mental disorders. The availability of national-level data on the prevalence and hospitalisation patterns of mental health problems is necessary for planning and enhancing clinical services that are provided to patients. This will enable an understanding of trends in the prevalence and hospitalisation of mental health disorders and, thus, shape the strategies to improve mental health. Nevertheless, no comprehensive national studies have been done that specifically investigate trends in hospital admission for mental health within all age groups in the UK. To address this gap, we aimed to analyse trends in hospital admissions due to mental health causes in the United Kingdom.

## 2. Methods

### 2.1. Data Sources and Study Population

As previously mentioned [9,11], we conducted an ecological study using publicly available data taken from the two main medical databases in England and Wales; the Hospital Episode Statistics (HES) database in England and the Patient Episode Database for Wales (PEDW) [20,21]. The HES database provides detailed information on hospital admissions associated with a wide range of health conditions in England and the PEDW provides similar information related to Wales. 

Hospital admission (HA) data was collected for the period between April 1999 and March 2019. The HES and PEDW databases contain HA data for all types of mental, behavioural, and neurodevelopmental disorders (MBNDs) for patients from all age categories: below 15 years, 15–59 years, 60–74 years, and 75 years and above. We identified MBND-related HA using the tenth version of the International Statistical Classification of Diseases (ICD10) system [22]. All diagnostic codes for MBNDs (F01–F99) were used to identify all hospital admissions related to various types of MBNDs in England and Wales. The HES and PEDW databases record all hospital admissions, outpatient visits, and accident and emergency attendances performed at all National Health Service (NHS) trusts and any independent sector funded by them. Data for hospital admissions in England and Wales is available from the years 1999/2000 onwards. HES and PEDW data are checked regularly to ensure their validity and accuracy. To calculate the annual hospital admission rate for MBNDs, we collected mid-year population data between 1999 and 2019 from the UK Office for National Statistics [23].

### 2.2. Data Analysis

Annual MBND-related HA rates with 95% confidence intervals (CIs) were calculated using the number of hospital admissions related to each type of MBND for each age group divided by the mid-year population of the same age group in the same year. The trend in hospital admissions was assessed using a Poisson model. A two-sided *p* < 0.05 was considered statistically significant. All analyses were performed using SPSS version 27 (IBM Corp., Armonk, NY, USA).

## 3. Results

The overall yearly number of mental, behavioural, and neurodevelopmental disorder hospital admissions with various causes increased by 17.7% from 242,335 in 1999 to 285,149 in 2019, representing an increase in hospital admission rate of 3.2% [from 464.78 (95% CI 462.93–466.62) in 1999 to 479.73 (95% CI 477.97–481.48) in 2019 per 100,000 persons; *p* = 0.065]. 

The most common MBNDs leading to hospital admission were mental and behavioural disorders due to psychoactive substance use, accompanied by mood [affective] disorders, mental disorders due to known physiological conditions, and schizophrenia, schizotypal, delusional, and other non-mood psychotic disorders, which accounted for 26.6%, 18.1%, 16.7%, and 16.5%, respectively (Table 1). 

Over the past two decades, the most notable rise in the rate of hospital admissions (87.3%) has been for mental disorders due to known physiological conditions. Moreover, the rate of hospitalisation related to mental and behavioural disorders due to psychoactive substance use, anxiety, dissociative, stress-related, somatoform and other nonpsychotic mental disorders, behavioural syndromes associated with physiological disturbances and physical factors, and disorders of adult personality and behaviour increased by 58.3%, 37.3%, 27.2%, and 26.7%, respectively. Conversely, the rate of hospital admissions due to intellectual disabilities, unspecified mental disorders, mood [affective] disorders, pervasive and specific developmental disorders, behavioural and emotional disorders with onset usually occurring in childhood and adolescence, and schizophrenia, schizotypal, delusional, and other non-mood psychotic disorders decreased by 95.9%, 87.6%, 48.5%, 19.4%, 14.3%, and 10.8%, respectively (Figure 1).

Regarding age group variation in MBND hospital admissions, the 15–59 years’ age group accounted for 65.1% of the entire number of such hospital admissions, followed by the 75 years and above age group, with 19.0%, the age group 60–74 years, with 12.7%, and then the age group below 15 years, with 3.3%. Age is not the only relevant factor concerning the increase in hospital admissions related to MBNDs (Figure 2). MBND hospital admission rates among patients aged below 15 years decreased by 57.5% [from 139.67 (95%CI 137.35–142.00) in 1999 to 59.42 (95%CI 57.96–60.88) in 2019 per 100,000 persons; *p* < 0.05]. The rate among patients aged 15–59 years decreased by 0.8% [from 493.45 (95%CI 491.00–495.91) in 1999 to 489.35 (95%CI 487.02–491.68) in 2019 per 100,000 persons; *p* = 0.228]. The rate among patients aged 60–74 years decreased by 3.0% [from 441.13 (95%CI 436.20–446.06) in 1999 to 428.09 (95%CI 423.88–432.30) in 2019 per 100,000 persons; *p* = 0.852]. The rate among patients aged 75 years and above increased by 26.9% [from 1099.60 (95%CI 1089.27–1109.93) in 1999 to 1395.92 (95%CI 1385.71–1406.12) per 100,000 persons in 2019; *p* < 0.001] (Figure 2).

Overall, 4,914,091 MBND hospital admission episodes were recorded in England and Wales during the duration of the study. Males contributed to 52.9% of the total number of MBND-related hospital admissions, accounting for 2,598,458 admissions with a mean of 129,922 per year. The MBND hospital admission rate for males increased by 8.1% [from 479.59 (95% CI 476.90–482.27) in 1999 to 518.30 (95% CI 515.71–520.90) in 2019 per 100,000 persons; *p* < 0.001]. The same rate for females increased by 0.3% [from 451.45 (95% CI 448.91–453.99) in 1999 to 452.77 (95% CI 450.37–455.17) in 2019 per 100,000 persons; *p* = 0.547] (Figure 3).

### 3.1. Mental, Behavioural and Neurodevelopmental Disorders Admission Rate by Gender

MBND hospital admission rates related to psychoactive substance use, schizophrenia, schizotypal, delusional and other non-mood psychotic disorders, intellectual disabilities, pervasive and specific developmental disorders, behavioural, and emotional disorders with onset usually occurring in childhood and adolescence, and unspecified mental disorders [12] were higher among males compared to females, while rates for mental disorders due to known physiological conditions, mood [affective] disorders, anxiety, dissociative, stress-related, somatoform and other nonpsychotic mental disorders, behavioural syndromes associated with physiological disturbances and physical factors, and disorders of adult personality and behaviour [12] were higher among females (Figure 4).

### 3.2. Mental, Behavioural and Neurodevelopmental Disorders Hospital Admission Rate by Age Group

Hospital admissions for mental disorders due to known physiological conditions, and mood [affective] disorders were seen to be directly related to age. Admissions due to anxiety, and dissociative, stress-related, somatoform, and other nonpsychotic mental disorders, and unspecified mental disorders were more common among the age groups of 75 years and above, 15–59 years, 60–74 years, and below 15 years, respectively. Hospital admissions due to psychoactive substance use, schizophrenia, schizotypal, delusional, and other non-mood psychotic disorders, and disorders of adult personality and behaviour were more common among the age groups 15–59 years, 60–74 years, 75 years and above, and below 15 years, respectively. Hospital admissions due to behavioural syndromes associated with physiological disturbances and physical factors were more common among the age groups 15–59 years, below 15 years, 60–74 years, and 75 years and above, respectively. Admissions due to pervasive and specific developmental disorders, and behavioural and emotional disorders with onset usually occurring in childhood and adolescence were more common among the age groups below 15 years, 15–59 years, 75 years and above, and 60–74 years, respectively. Nevertheless, intellectual disabilities hospital admissions were inversely related to age, being more common among the age group below 15 years (Figure 5).

## 4. Discussion

To our knowledge, this is the first study to explore the trend in MBND hospital admissions in England and Wales over a 20-year period (1999–2019). According to mental health statistics for the UK and worldwide, mental health problems are one of the leading causes of the overall disease burden worldwide [2]. A key driver of disability worldwide is attributed to mental health and behavioural disorders such as depression, anxiety, and drug abuse. This resulted in more than 40 million years of disability among patients in the category of 20–29 years old in 2010 [24]. Furthermore, according to a study by Whiteford et al. on the global burden of disease attributable to mental and substance use disorders, major depression was considered to be the second leading cause of disability and a primary contributor to the burden of suicide and ischaemic heart disease [25].

This study has identified significant increases in hospital admissions for MBNDs in England and Wales, between 1999 and 2019, for most subcategories of mental disorders. The findings have revealed that mental and behavioural disorders due to psychoactive substance use (ICD_10_ code F10-19) contributed to 26.6% of MBND hospital admissions and demonstrated a striking increase in admission rates over the two decades (from 87.16 to 137.97 admissions in a 100,000 population; *p* < 0.001) in 1999/2002 and 2018/2019, respectively (see Figure 1). Under this category, mental disorders are based on substance use in terms of the use of alcohol (F10), opioid and cannabis (F11 and F12, respectively), sedatives, hypnotics, or anxiolytics (F13), cocaine (F14), other stimulants (F15), hallucinogens (F16), nicotine dependence (F17), inhalants (F18) and other substance-related disorders (F19) [22]. The findings in this study follow studies in other countries. In a study by Fløvig et al. (2009) in Norway, substance use accounted for 81.9% of mental health hospital admissions (186/227). Details of the most commonly used substances revealed that benzodiazepines accounted for 52.4% of admissions, followed by alcohol at 41.4% [26]. Research by Hayes et al. (2011) in the UK revealed that alcohol and opioid use disorders were the most prevalent, accounting for 45.4% and 44.2% of the study population, respectively. Unfortunately, this increase in substance use resulted in hospital admissions becoming a significant burden [27]. A recent study by Al-Daghastani and Naser examined the trend of hospital admissions related to poisoning by, adverse effects of, and underdosing of, psychotropic drugs [8]. This study found that during the past two decades there was an increase in the admission rate of 20.0% [8]. Antidepressants, tricyclic and tetracyclic antidepressants, antipsychotics and neuroleptics, and psychostimulants with abuse potential were the most frequently cited causes of hospital admissions related to poisoning by psychotropic medications, accounting for 48.9%, 20.9%, 13.4%, and 8.3%, respectively [8]. The causes for these admissions were different, which were accidental (unintentional), intentional self-harm, assault as an adverse effect, and under-dosing. One possible reason for the increase in this type of admission is the increase in the prescription rate of psychotropic medications [28]. A recent study in the UK reported that the prescription rate of central nervous system medications increased by 54.1% during the period between 2004 and 2019 [29]. Antidepressant medications accounted for 28.0% of the total number of medications prescribed for central nervous system conditions [29]. Besides, a previous study in the United States has found that between 2003 and 2017, the mortality rate from psychostimulant addiction and poisoning rose by a third [30]. All of these increases in drug-related intoxication and poisoning produced by various psychotropic medications increased the demand for hospital admission. [31]. Some psychiatric medications increase the risk of suicidal ideation, which can lead to self-poisoning and attempts at self-harm [32]. In response to that, the UK government identified alcohol misuse as a main public health concern, with 1.6 million individuals dependent on alcohol in 2007. The UK government set out a strategy (reduction in drug abuse) and set out how to tackle the burden of this misuse [7].

Interestingly, despite the high hospitalisation rate for mood [affective] disorders, such as depressive and bipolar disorders, the hospitalisation rate was reduced by more than half in this category (from 112.10 to 57.75 admissions per 100,000; *p* < 0.001) in 1999/2000 and 2018/2019, respectively). Although the year-on-year changes are relatively small, the sustained trends over many years have led to a significant reduction in the hospitalisation rate of such a long-standing mental health condition (as highlighted in Figure 1). An earlier study, conducted between 1994 and 1999, looked at the effects of new antidepressant medications on patient adherence and the risk of a cycle of acceleration in affective disorders, and thus hospitalization. The findings of this study have revealed that severe depressive and bipolar disorders, including hospitalisation have remained roughly the same, even with the use of new treatments [33]. However, a more recent study in 2017 showed that the effectiveness of medication used for the treatment of bipolar disorder resulted in reduced rates of hospital admission [34]. Such results tally with the findings of this study.

Gender differences in mental disorders are among the most fascinating and well-known findings in mental health [35]. The findings of this research reveal a widening gender gap with, overall, males accounting for an increasingly greater proportion of hospital admissions. Similar findings were reported earlier in the UK during 1999–2000 [36]. Interestingly, this has not translated to a similar trend when subcategories of MBNDs are stratified. Admission rates were significantly higher among females in disorders with known psychological conditions (F01–F09), mood (affective) disorders (F30–F39), anxiety, dissociative and stress-related disorders (F40–F48), behavioural syndromes (F50–F59), as well as adult personality and behavioural disorders (F60–F69), where any gender gap in the later years increased, particularly in behavioural syndromes (F50–F59) and adult personality and behavioural disorders (F60–F69). These findings are consistent with previous research indicating that females have a higher prevalence of mood or anxiety disorders than males [35,37,38]. We found a consistent and persisting increase in female MBND hospital admissions related to behavioural syndromes associated with psychological disturbances and physical factors (F50–F59), where the female hospitalisation trend continued to rise from 9.06 hospital episodes per 100,000 in 1999/2000 to 18.71 hospital episodes per 100,000 in 2018/2019. In the same category, the hospitalisation rate for male patients dropped from 9.06 to 3.83 hospital episodes in 1999/2000 and 2018/2019, respectively. This category is related to several disorders, such as eating disorders (F50) and sleeping disorders (F51) [22]. Several studies have reported that the prevalence rate of eating disorders is higher in females than in males. For example, almost 90% of individuals diagnosed with bulimia nervosa are female [39,40]. Such a variation in gender-related hospitalisation trends might provoke policymakers to set strategies that are gender-specific to reduce such trends.

Our findings were further stratified into age groups, as depicted in Figure 2. The results showed that the only category that demonstrated an increase in the hospitalisation rate was that of patients exceeding 75 years of age. This could be partially attributed to the increase in hospitalisation related to mental disorders due to the known physiological conditions that are the main contributors to dementia. The prevalence rate of dementia increases with age. When the age-related admissions due to mental MBNDs were stratified by diagnosis, trends showed that for mental disorders due to known physiological conditions, there was a significant increase in admission rates among the age group >75 years. Similar results were reported in other studies in Australia in 2009 [41]. However, for mental and behavioural disorders due to psychoactive substance use, the age categories 15–59 followed by the category 60–74 years showed an increase in admission rates over the two decades. Our findings reflect a report by the NHS-UK for the period 2015–2017, which disclosed that about 33% of patients admitted for drug-related mental and behavioural disorders were between the ages of 25 to 34, followed by the 35 to 44 age group, which accounted for 26% [42]. In addition, this was also aligned with the findings of a recent study in the UK, which found that the age group 15–59 years accounted for the majority of hospital admissions related to psychotropic medication poisoning [8]. A previous study by Paulose-Ram et al. found that this age group has a high utilization rate of psychotropic medications [21]. As for paediatrics, the only category that demonstrated an increase in admission rates was behavioural syndromes associated with physiological disturbances, while other categories showed consistent or reduced admission rates. In comparison with other countries, a USA-based study showed that almost 10% of paediatric hospitalisations were for a primary mental health diagnosis. However, the highest number of admissions among children for depression accounted for 44.1% of all mental health hospital admissions, costing $1.33 billion, followed by bipolar disorder, accounting for more than $700 million, with an admission rate of 18.1% [43].

Our findings emphasise the need for further research in the UK for mental health disorders that addresses the striking increase in admission rates for specific ICD 10 subcategories, genders, and age groups. The causes behind this ought to be investigated. These results have policy relevance in the context of the implementation of the new national mental health strategy in England (No health without mental health: A cross-government mental health outcomes strategy for people of all ages) [7].

To our knowledge, this is the first study in the UK to address the trends in hospitalisation over two decades due to mental health disorders. Therefore, a key strength of this study is its large size and long duration, whereby it includes all admission records of patients with a diagnosis of MBNDs in the study period and is, consequently, nationally representative. Such study characteristics support the generalisability of the findings compared with previous studies in the UK, which have been limited to specific cases, were of shorter duration, or conducted at limited sites. There are some limitations to this study arising from the use of HSE data as routinely collected data, including information from hospital records that has not been collected for research purposes. The data publically available from the two medical databases are on the population level and not on the individual level. This restricted our ability to adjust for important confounding variables and extract information related to patients’ demographic characteristics, comorbidities, and drug use history. Consequently, we were unable to identify other risk factors that could be associated with the development of MBNDs. Other limitations include a lack of information on gender at the age-group level, rural versus urban residence, and ethnicity.

Future studies on the individual patient level are warranted to identify other important risk factors of MBND hospital admissions. In addition, they should identify high-risk medications associated with these types of admissions.

## 5. Conclusions

Hospital admission rates have increased for most types of MBNDs, specifically for mental and behavioural disorders due to psychoactive substance use. Our study demonstrates an evident variation in MBNDs based on age and gender. Hospital admission rates were higher among males, the middle-aged, and elderly populations. Further observational studies are needed to identify other factors associated with increased hospital admission rates related to MBNDs among the general population and, specifically, for the young population (aged 15–59 years), the elderly (75 years and above), and males.

## Figures and Tables

**Figure 1 healthcare-10-02191-f001:**
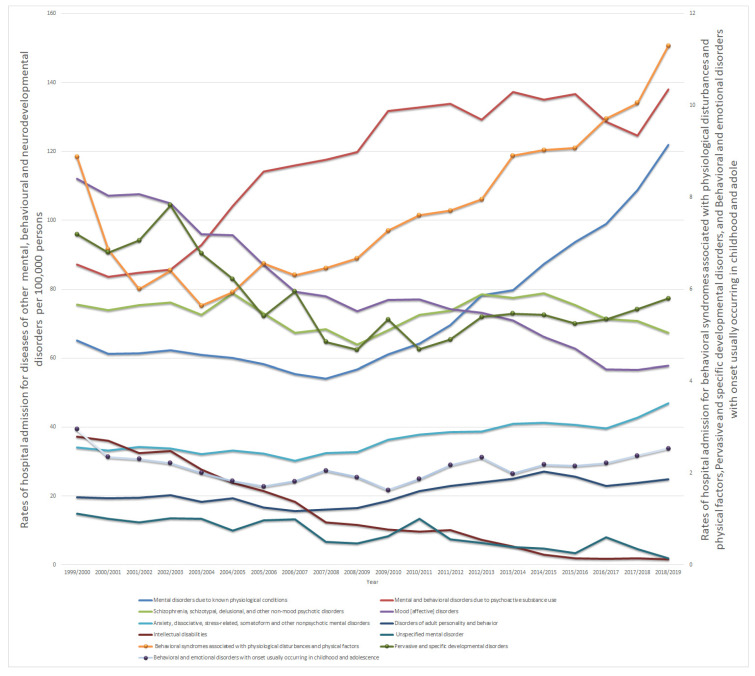
Hospital admission rates due to mental, behavioural, and neurodevelopmental disorders in England and Wales, stratified by type, between 1999 and 2019.

**Figure 2 healthcare-10-02191-f002:**
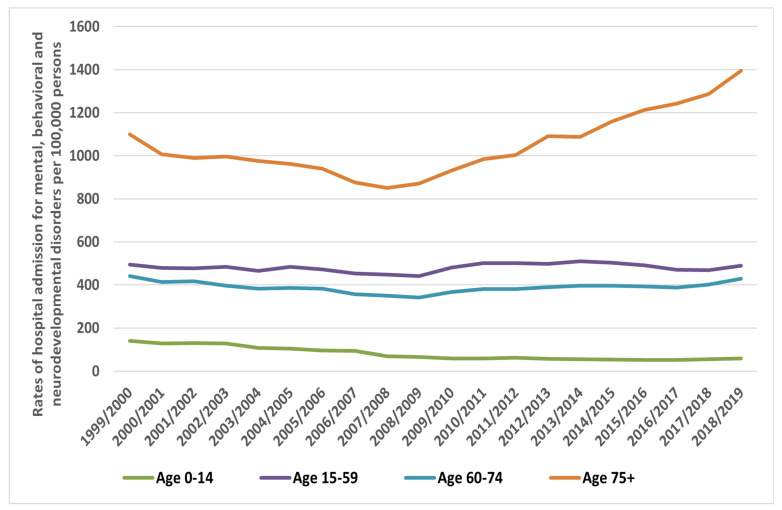
Rates of hospital admissions for all MBNDs in England and Wales stratified by age group.

**Figure 3 healthcare-10-02191-f003:**
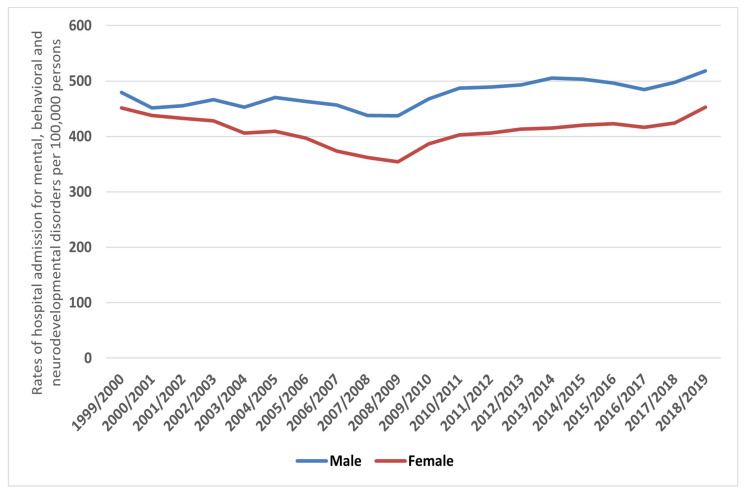
Rates of hospital admissions for all MBNDs in England and Wales stratified by gender.

**Figure 4 healthcare-10-02191-f004:**
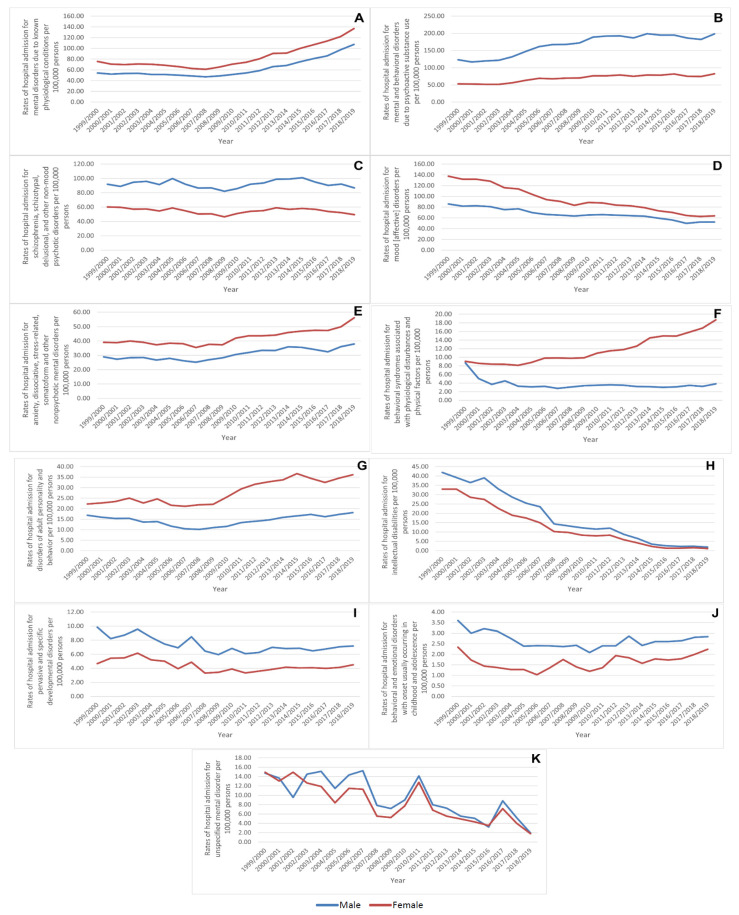
Hospital admission rates for MBNDs in England and Wales stratified by gender. ((**A**) Mental disorders due to known physiological conditions, (**B**) Mental and behavioral disorders due to psychoactive substance use, (**C**) Schizophrenia, schizotypal, delusional, and other non-mood psychotic disorders, (**D**) Mood [affective] disorders, (**E**) Anxiety, dissociative, stress-related, somatoform and other nonpsychotic mental disorders, (**F**) Behavioral syndromes associated with physiological disturbances and physical factors, (**G**) Disorders of adult personality and behavior, (**H**) Intellectual disabilities, (**I**) Pervasive and specific developmental disorders, (**J**) Behavioral and emotional disorders with onset usually occurring in childhood and adolescence, and (**K**) Unspecified mental disorder).

**Figure 5 healthcare-10-02191-f005:**
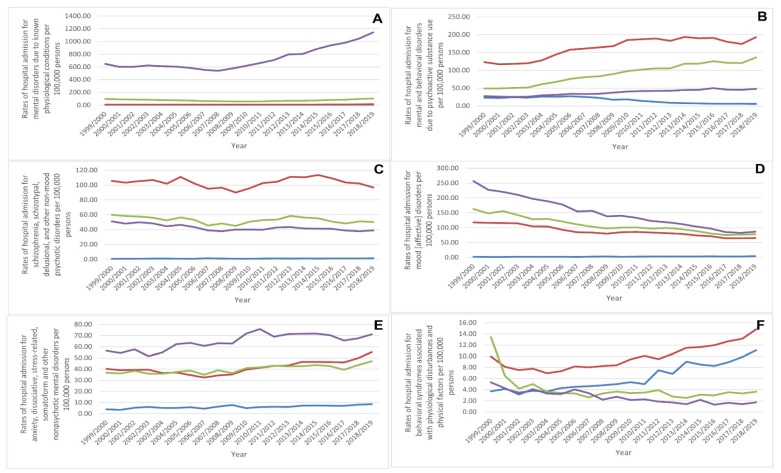
Hospital admission rates for MBNDs in England and Wales stratified by age group. ((**A**) Mental disorders due to known physiological conditions, (**B**) Mental and behavioral disorders due to psychoactive substance use, (**C**) Schizophrenia, schizotypal, delusional, and other non-mood psychotic disorders, (**D**) Mood [affective] disorders, (**E**) Anxiety, dissociative, stress-related, somatoform and other nonpsychotic mental disorders, (**F**) Behavioral syndromes associated with physiological disturbances and physical factors, (**G**) Disorders of adult personality and behavior, (**H**) Intellectual disabilities, (**I**) Pervasive and specific developmental disorders, (**J**) Behavioral and emotional disorders with onset usually occurring in childhood and adolescence, and (**K**) Unspecified mental disorder).

**Table 1 healthcare-10-02191-t001:** Percentage of mental, behavioural, and neurodevelopmental disorder admissions from the total number of admissions per ICD code.

ICD_10_ Code	Description	Percentage from Total Number of Admissions
**F01–F09**	Mental disorders due to known physiological conditions	16.7%
**F10–F19**	Mental and behavioral disorders due to psychoactive substance use	26.6%
**F20–F29**	Schizophrenia, schizotypal, delusional, and other non-mood psychotic disorders	16.5%
**F30–F39**	Mood [affective] disorders	18.1%
**F40–F48**	Anxiety, dissociative, stress-related, somatoform and other nonpsychotic mental disorders	8.3%
**F50–F59**	Behavioral syndromes associated with physiological disturbances and physical factors	1.8%
**F60–F69**	Disorders of adult personality and behavior	4.8%
**F70–F79**	Intellectual disabilities	3.4%
**F80–F89**	Pervasive and specific developmental disorders	1.3%
**F90–F98**	Behavioral and emotional disorders with onset usually occurring in childhood and adolescence	0.5%
**F99–F99**	Unspecified mental disorder	2.0%

ICD: International Statistical Classification of Diseases system.

## Data Availability

Publicly available datasets were analyzed in this study. This data can be found here: http://content.digital.nhs.uk/hes (accessed on 13 May 2021), http://www.infoandstats.wales.nhs.uk/page.cfm?pid=41010&orgid=869 (accessed on 13 May 2021).

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
