# Peer review of "Trends in Hospital Admissions for Mental, Behavioural and Neurodevelopmental Disorders in England and Wales between 1999 and 2019: An Ecological Study"

_healthcare, 2022, doi:10.3390/healthcare10112191_

Round 1
Reviewer 1 Report
The research aims to identify how the trends in hospital admissions due to mental disorders in England and Wales has changed during a 20 years timespan. The topic is interesting because by knowing the different trends, governments can implement containment measures to counter the phenomenon.
Although not a completely original topic, the study is the only one that focuses on the UK national scenario, analyzing a huge and complete amount of data, encompassing up to 20 years.
The contribution is well written, the text is clear and the structure is solid.
The conclusion, according to which the number of hospital admissions for mental disorders has increased, but showing different trends based on age and gender, is consistent with the analysis of the data.
The authors state that further studies are needed to identify other factors associated with increased hospital admission rates.
Discussions should be broadened.
Author Response
- First of all, we would like to thank the reviewer for the time and efforts in reviewing our manuscript. Your comments and feedback are highly appreciated.
The research aims to identify how the trends in hospital admissions due to mental disorders in England and Wales has changed during a 20 years timespan. The topic is interesting because by knowing the different trends, governments can implement containment measures to counter the phenomenon. Although not a completely original topic, the study is the only one that focuses on the UK national scenario, analyzing a huge and complete amount of data, encompassing up to 20 years.
The contribution is well written, the text is clear and the structure is solid.
The conclusion, according to which the number of hospital admissions for mental disorders has increased, but showing different trends based on age and gender, is consistent with the analysis of the data.
The authors state that further studies are needed to identify other factors associated with increased hospital admission rates.
Discussions should be broadened.
- Thank you for your comment, we have now broadened the discussion based on the reviewer comment, see lines 228-247 and lines 302-306.
Reviewer 2 Report
This study investigated the trends in hospital admission for mental, behavioral and neurodevelopmental disorders in England and Wales between 1999 and 2019, which is interesting theme for periodic changes of mental health issue. However, there are several questions remained.
1. Why do you investigate dealing with database of England and Wales, because the authors are from Jordan, Saudi Arabia, or Kuwait.
2. Abstract, line 20: there seems missing year in the sentence.
3. Abstract, line 22: there is no result regarding age, but the conclusion shows age-related one.
4. Introduction: The authors should explain why mental, behavioral and neurodeveloping disorders were evaluated together. There seems no association of neurodeveloping disorder with others. The neurodeveloping disorder is somewhat congenital. It is better that the more clear explanation should be described in the introduction.
5. The conclusions need to rewrite so that they explain your outcomes in detail.
Author Response
- First of all, we would like to thank the reviewer for the time and efforts in reviewing our manuscript. Your comments and feedback are highly appreciated.
This study investigated the trends in hospital admission for mental, behavioral and neurodevelopmental disorders in England and Wales between 1999 and 2019, which is interesting theme for periodic changes of mental health issue. However, there are several questions remained.
- Why do you investigate dealing with database of England and Wales, because the authors are from Jordan, Saudi Arabia, or Kuwait.
- Thank you for this comment. The first author Dr. Abdallah Y Naser is an epidemiologist who did his PhD studies at University College London, which is in the United Kingdom. These data are publically available data that are of huge value and not available in the Middle East region. There is no electronic medical databased that connect all hospital data (private and public) in Jordan, Saudi Arabia, or Kuwait. Besides, exploring epidemiological data help us in generating hypothesis and research question that can be investigated further on smaller scale in Jordan, Saudi Arabia and Kuwait. However, it’s worth mentioning that our co-author Dr. Eman Zmaily Dahmash is from Kingston University. I have now amended her affiliation details.
- Abstract, line 20: there seems missing year in the sentence.
- Thank you for this comment. We have now addressed this point, see lines 21-23.
- Abstract, line 22: there is no result regarding age, but the conclusion shows age-related one.
- Thank you for this comment. We have now addressed this point, see lines 23-25.
- Introduction: The authors should explain why mental, behavioral and neurodeveloping disorders were evaluated together. There seems no association of neurodeveloping disorder with others. The neurodeveloping disorder is somewhat congenital. It is better that the more clear explanation should be described in the introduction.
- Thank you for this comment. Mental, behavioral and neurodeveloping disorders are grouped together in the International Classification of Diseases (ICD)-10 classification system. In addition, we were interested in exploring the admissions trends for them. However, we stratified them in the results section and presented them separately in figures 4 and 5. We have now added the following statements for the introduction section “A clinically significant disturbance in a person's cognition, emotional control, or behavior indicative of a problem with the psychological, biological, or developmental processes underlying mental and behavioral functioning characterizes mental, behav-ioral, and neurodevelopmental disorders”, see lines 63-66.
- The conclusions need to rewrite so that they explain your outcomes in detail.
- Thank you for this comment. We have now highlighted further the key findings in our conclusion, see lines 342-346.
Reviewer 3 Report
The Manuscript ,,trends in hospital admission for mental, behavioural and neurodevelomental disororders in england and Wales between 199 and 2019: an ecological study'' i s very interesting.
I have only two suggestions:
1. for me the division according the age for age 0-14, 15-59, 60-74 and age 75+ is not good. I suggest to divide to age 0-18; 18-30; 31-59, 60-75; 75+
2. I suggest to change the color in the Figures because it is difficult to identify the groups
Author Response
- First of all, we would like to thank the reviewer for the time and efforts in reviewing our manuscript. Your comments and feedback are highly appreciated.
The Manuscript,, trends in hospital admission for mental, behavioural and neurodevelomental disororders in england and Wales between 199 and 2019: an ecological study'' i s very interesting.
I have only two suggestions:
- for me the division according the age for age 0-14, 15-59, 60-74 and age 75+ is not good. I suggest to divide to age 0-18; 18-30; 31-59, 60-75; 75+
- Thank you for this comment. Unfortunately, the two medical databases reported the data stratified into four main age groups, which are the ones that we used to present the data. We don’t have the data in the format that the reviewers asked for.
- I suggest to change the color in the Figures because it is difficult to identify the groups
- Thank you for this comment. We have now addressed the reviewer comment, see Figure 1. In addition, upon the production of the article before publication, this will be edited by the production team of the journal.
Reviewer 4 Report
The following comments need to be addressed for consideration for publication.
There is no mention of other comorbidities that may lead to the increase in hospitalization. Authors should mention in a graph the rate of hospitalization by types of drugs used. Were there any other factors related to the increase in hospitalization?
In Fig 2, are all the types of disorders combined and reflected on the graph.
Add the limitations for your study and some future research directions.
Add a statistical methods section to explain the methods used for the analysis.
Is there any kind of model used to identify the factors responsible for hospital admission? Only, descriptive statistics are provided.
Author Response
- First of all, we would like to thank the reviewer for the time and efforts in reviewing our manuscript. Your comments and feedback are highly appreciated.
The following comments need to be addressed for consideration for publication.
There is no mention of other comorbidities that may lead to the increase in hospitalization. Authors should mention in a graph the rate of hospitalization by types of drugs used. Were there any other factors related to the increase in hospitalization?
- Thank you for this comment. Unfortunately, the available data are on the population level not on the individual level, therefore, we do not have data related to patients’ comorbidities or drug use history associated with these hospital admissions. However, we have no discussed this point in the discussion section 228-247 and lines 302-306. Besides, we have now added this point to the limitation section based on the reviewer comment, see lines 329-332.
In Fig 2, are all the types of disorders combined and reflected on the graph.
- Thank you for this comment. Yes, we presented hospital admissions stratified by age groups for all types of admissions, then, we stratified it by indication in figure 5.
Add the limitations for your study and some future research directions.
- Thank you for this comment. We have now addressed the reviewer comment and added limitations for study and future research directions, see lines 336-338.
Add a statistical methods section to explain the methods used for the analysis.
- The statistical method section is available in lines 99-105 under the sub-heading “Data analysis”.
Is there any kind of model used to identify the factors responsible for hospital admission? Only, descriptive statistics are provided.
- Thank you for this comment. Unfortunately, the available data are on the population level not on the individual level, therefore, we do not have data related to patients’ comorbidities, drug use history, or other risk factor associated with these hospital admissions. We have now added this point to the limitations of the study, see lines 329-335.
Round 2
Reviewer 2 Report
I think well revision in this form. I hope the authors contribute to your country's public health organization.
Reviewer 4 Report
No further comments. The authors have addressed them.